# Yuanhuacine Is a Potent and Selective Inhibitor of the Basal-Like 2 Subtype of Triple Negative Breast Cancer with Immunogenic Potential

**DOI:** 10.3390/cancers13112834

**Published:** 2021-06-07

**Authors:** Charles S. Fermaintt, Thilini Peramuna, Shengxin Cai, Leila Takahashi-Ruiz, Jacob Nathaniel Essif, Corena V. Grant, Barry R. O’Keefe, Susan L. Mooberry, Robert H. Cichewicz, April L. Risinger

**Affiliations:** 1Department of Pharmacology, Mays Cancer Center, University of Texas Health Science Center San Antonio, San Antonio, TX 78229, USA; Fermaintt@uthscsa.edu (C.S.F.); takahashil@livemail.uthscsa.edu (L.T.-R.); essif@livemail.uthscsa.edu (J.N.E.); corena.grant@osumc.edu (C.V.G.); mooberry@uthscsa.edu (S.L.M.); 2Department of Chemistry and Biochemistry and Natural Products Discovery Group, University of Oklahoma, Norman, OK 73019, USA; peramunu.g.peramuna-1@ou.edu (T.P.); shengxincai2012@gmail.com (S.C.); rhcichewicz@ou.edu (R.H.C.); 3Natural Products Branch, Developmental Therapeutics Program, Division of Cancer Treatment and Diagnosis and Molecular Targets Program, Center for Cancer Research, National Cancer Institute, Frederick, MD 21702, USA; okeefeba@mail.nih.gov

**Keywords:** triple negative breast cancer, basal like 2, targeted therapy, immunotherapy, daphnane type diterpenoids, protein kinase C, natural products, pharmacology, traditional Chinese herbal medicine

## Abstract

**Simple Summary:**

Most breast cancers express the estrogen, progesterone, and/or HER2 receptors and patients are treated with inhibitors targeting these receptors. Triple negative breast cancers (TNBCs) lack these receptors and thus patients with TNBC do not benefit from existing targeted therapies. There is a continued search for effective targets for the treatment of these heterogeneous tumors. We identified a class of plant-derived natural products, including yuanhuacine, that selectively kill cells that represent a molecularly defined subtype of TNBC. These compounds also promote expression of an immunological profile that is beneficial for engaging the immune system, which could provide an added benefit in TNBC. The mechanism of action of both the TNBC selectivity and immunological phenotypes is associated with activation of protein kinase C. Yuanhuacine has potent antitumor efficacy in a mouse model of TNBC, identifying a new therapeutic target for the treatment of this deadly disease.

**Abstract:**

The heterogeneity of triple negative breast cancer (TNBC) has led to efforts to further subtype this disease with the hope of identifying new molecular liabilities and drug targets. Furthermore, the finding that TNBC is the most inherently immunogenic type of breast cancer provides the potential for effective treatment with immune checkpoint inhibitors and immune adjuvants. Thus, we devised a dual screen to identify compounds from natural product extracts with TNBC subtype selectivity that also promote the expression of cytokines associated with antitumor immunity. These efforts led to the identification of yuanhuacine (**1**) as a potent and highly selective inhibitor of the basal-like 2 (BL2) subtype of TNBC that also promoted an antitumor associated cytokine signature in immune cells. The mechanism of action of yuanhuacine for both phenotypes depends on activation of protein kinase C (PKC), defining a novel target for the treatment of this clinical TNBC subtype. Yuanhuacine showed potent antitumor efficacy in animals bearing BL2 tumors further demonstrating that PKC could function as a potential pharmacological target for the treatment of the BL2 subtype of TNBC.

## 1. Introduction

Breast cancer remains the second leading cause of cancer-related deaths in women worldwide [1]. Triple negative breast cancer (TNBC) is defined by the absence of estrogen, progesterone, and/or HER2 receptors. While highly effective drugs targeting these receptors have been developed, they are not effective against TNBC [2,3,4]. Although many patients respond positively to the standard of care chemotherapy regimens, more than half of all TNBC patients do not have a durable response [5,6]. Thus, the identification of new targets and drug leads for the treatment of TNBC will be required to increase the long-term survival of these patients. These efforts are complicated by the highly heterogeneous nature of TNBC [7]. Gene expression profiling of TNBC tumors by Lehmann and Bauer et al. originally identified six molecularly distinct subtypes with representative TNBC cell lines assigned to each [8]. With more refined laser dissection techniques, the subtypes were consolidated into basal-like 1 and 2 (BL1 and BL2), mesenchymal (M) and luminal androgen receptor (LAR) subtypes [9,10]. Our group has previously utilized this molecular subtyping to identify natural products with selective activity against cell lines representing the BL1, M, and LAR subtypes of TNBC to inform on unanticipated druggable targets [11,12,13,14,15].

In addition to the development of targeted inhibitors of the drivers of specific TNBC subtypes, the use of immune checkpoint inhibitors has also demonstrated potential for improving TNBC patient outcomes due to the relatively high degree of immunogenicity of most TNBCs compared to other types of breast cancer [16]. Evaluations of combinations of chemotherapeutics, including nab-paclitaxel or eribulin, with immune checkpoint inhibitors in TNBC suggest this strategy could lead to improved response, particularly in patients with an immunologically favorable tumor microenvironment [17,18]. We hypothesize that agents with selectivity for a molecularly defined TNBC subtype that also increase tumor immunogenicity could prove particularly beneficial in the treatment of TNBC.

Herein, we describe the identification of a class of daphnane type diterpenoids, including yuanhuacine, that demonstrate a striking, over 100-fold, selectivity for cells representing the BL2 subtype of TNBC, which is characterized by the enrichment of molecular pathways involved in growth factor signaling, glycolysis and gluconeogenesis [8]. Additionally, these compounds promote the differentiation of THP-1 monocytes into adherent myeloid cells that express cytokines associated with an antitumor immunological response. Mechanistic characterization revealed that both the BL2 selective cytotoxicity and immunogenicity occur through activation of protein kinase C (PKC). The antitumor efficacy of yuanhuacine against BL2 subtype HCC1806 tumors highlights the unanticipated role of PKC as a druggable target specifically for the treatment of this molecular subtype of TNBC.

## 2. Materials and Methods

### 2.1. Cell Lines

BT-549 cells were obtained from the Georgetown University Lombardi Comprehensive Cancer Center (Washington, DC, USA). All other cell lines were purchased from the American Type Culture Collection (ATCC, Manassas, VA, USA). All TNBC lines were authenticated by STR-based profiling (Genetica DNA Laboratories, Cincinnati, OH, USA). MDA-MB-231 and MDA-MB-453 cells were maintained in IMEM medium (Gibco, Grand Island, NY, USA) supplemented with 10% FBS (Corning, Corning, NY, USA) and 50 μg/mL gentamycin (LifeTechnologies, Carlsbad, CA, USA). THP-1, HCC1937, HCC1806, HCC70 and BT-549 cells were maintained in RPMI 1640 medium (Corning) supplemented with 10% FBS and 50 μg/mL gentamycin. RAW 264.7 cells were maintained in DMEM medium (Gibco) supplemented with 10% FBS and 50 μg/mL gentamycin. All cells were grown inside an incubator kept at 37 °C and 5% CO_2_ and routinely tested negative for mycoplasma contamination.

### 2.2. Reagents

Drugs, ligands and inhibitors used in this study include etoposide (Sigma Aldrich, St. Louis, MO, USA), SN-38 (Cayman Chemical, Ann Arbor, MI, USA), gemcitabine (Sigma Aldrich), carbonyl cyanide *m*-chlorophenylhydrazone (CCCP, Sigma Aldrich), phorbol 12-myristate 13-acetate (PMA, Sigma Aldrich), bryostatin 1 (Tocris, Bristol, UK), Ro 31-8220 (Tocris), TPCA-1 (Sigma Aldrich), recombinant human IL-4 (PharMingen, Franklin Lakes, NJ, USA) and paclitaxel (LC Labs, Woburn, MA, USA). All drugs and inhibitors were dissolved in DMSO (Fisher, Waltham, MA, USA) except recombinant human IL-4 which was dissolved in water and paclitaxel which was dissolved 50:50 Kolliphor/ethanol.

### 2.3. Isolation of 1 and Other Daphnane Type Diterpenoids

The bioactive N117965 crude plant extract was derived from a collection on 15 May 1995 in Taylor County (FL, USA). The material (collection #WBA-2875) was collected and identified by Spjut and Gilliland as a possible *Myrica inodora* Batr. (Myricaceae) under contract to the Natural Products Branch of the NCI. The plant material (stems and leaves) was extracted (50% MeOH/50% CHCl_2_) by the NCI Natural Products Branch (Frederick MD, USA) to provide the material used for these studies. To identify the bioactive components, 5.0 g of this extract was fractionated over a vacuum-liquid-chromatography HP20ss column using a step gradient of MeOH-H_2_O (30:70, 50:50, 70:30, 90:10 and 100:0) and washed with CH_2_Cl_2_-MeOH (50:50). The fourth fraction (MeOH-H_2_O 90:10) was further fractionated by C_18_ preparative HPLC (MeCN-H_2_O 70:30) followed by semi-preparative HPLC using biphenyl column, to obtain 1 (0.8 mg). Additional 1 for in vivo studies and compounds **2**–**8** were obtained from 500 g of a dried *Daphne genkwa* sample purchased from a traditional medicine vendor (colorfullife051) based in China. The plant was received as dried material that had been finely chopped and crumbled precluding visual authentication of the sample. However, the natural products identified from the plant material were consistent with it having originated from a plant in the genus *Daphne*. The dried sample was dissolved in methanol to obtain 47 g of extract. The partitioning of the crude extract was carried out with ethyl acetate and water to obtain 18 g of organic extract. The organic extract was fractionated over a vacuum-liquid chromatography HP20ss column, eluted with a MeOH-H_2_O step gradient (30:70, 50:50, 70:30, 90:10, and 100:0) and washed with CH_2_Cl_2_-MeOH (50:50). The fifth fraction, 100% MeOH was further fractionated by C_18_ preparative HPLC (MeCN-H_2_O 70:30) followed by semi-preparative HPLC using biphenyl column and pentafluorophenyl columns to obtain 1 (8.5 mg), 2 (11.8 mg), 3 (2.5 mg), 4 (5.1 mg), 5 (1.3 mg), 6 (3.7 mg), 7 (2.0 mg) and 8 (1.0 mg). The structures of the compounds were determined by dereplication procedures that consisted of comparisons of experimentally derived NMR, MS, and other spectroscopic data to published data on these compounds [19,20,21,22].

### 2.4. Sulforhodamine B and Caspase 3/7 Viability Assays

The sulforhodamine B (SRB) assay was used to evaluate the antiproliferative and cytotoxic potencies of compounds and extracts on adherent TNBC cell lines as previously described [23]. Briefly, cells were plated in 96-well plates and allowed to adhere overnight. The cells were subsequently treated with indicated compounds and extracts for 48 h after which cells were fixed, stained, and absorbance acquired using a SpectraMax plate reader (Molecular Devices, San Jose, CA, USA). Each concentration was compared to the 0.5% DMSO vehicle treated cells to determine the effect on cellular density. The concentration that caused a 50% decrease in cell density was determined by non-linear regression analysis using GraphPad Prism 8 (GraphPad software, San Diego, CA, USA) and defined as the IC_50_. For PKC inhibition experiments, cells were pretreated with Ro 31-8220 for 4 h followed by treatment with **1** for 48 h with the inhibitor present. Data are representative of an average of 2–4 independent experiments ±SEM. The viability of the non-adherent THP-1 cells was evaluated by incubation with 50 μL/mL caspase 3/7 green reagent (Thermo-Fisher, Waltham, MA, USA) solution at 37 °C for 30 min. Cells were subsequently rinsed with PBS and analyzed for caspase cleavage using a Guava Muse Cell Analyzer. The data are expressed as percentage of viability and error bars represent the SEM from biological duplicates.

### 2.5. THP-1 Differentiation Assay

A variation of the traditional sulforhodamine B (SRB) assay was used to quantify THP-1 differentiation as an immune activation readout. Non-adherent THP-1 cells were plated at a density of 25,000 cells per well in a 96-well plate. Cells were subsequently treated with 0.5% DMSO vehicle control (negative control), 100 nM PMA (positive control) or compounds/extracts for 24 h. Non-adherent THP-1 cells were discarded and those that became adherent upon differentiation were fixed, stained and absorbance determined as described above for the SRB assay [23]. The concentration of extract or compound that caused 50% increase in differentiation as compared to DMSO control (0%) and PMA positive control (100%) was determined by non-linear regression analysis using GraphPad Prism 8 and defined as the EC_50_. Data are representative of the average of 2–4 independent experiments ±SEM.

### 2.6. Quantitative Real Time-PCR

RNA was extracted from THP-1 or RAW264.7 cells after indicated treatments using TRIzol (Ambion, Austin, TX, USA) as per manufacturer’s instructions. cDNA was synthesized with iScript cDNA synthesis kit (Bio-Rad, Hercules, CA, USA) and analyzed on a CFX qRT-PCR machine using iTaq Universal SYBR Green Supermix (Bio-Rad). mRNA fold change was calculated using the 2^−ΔΔCt^ method where GAPDH was used as the control. The utility of GAPDH as a control was confirmed based on the consistency of raw Ct values among the conditions evaluated for a single cell line. All treated and stimulated samples were compared to their corresponding vehicle control, which was set to a relative value of 1. All experiments were performed as two technical replicates with a cutoff of variability of less than 1 Ct within each of two independent biological replicates. The range of the biological replicate values is represented in every bar graph. Data were subject to unpaired *t*-test, ordinary one-way ANOVA or two-way ANOVA for statistical analysis as described in figure legends. All primers were ordered from Sigma Aldrich and can be found in Appendix A.

### 2.7. Athymic Nude Mice Antitumor Xenograft Studies

Female athymic 5–6 weeks old nude mice (ENVIGO, Indianapolis, IN, USA) were injected bilaterally with HCC1806 tumor fragments into each flank. When tumors reached an approximate volume of 100 mm^3^, animals were matched into three groups based on tumor size and take-rate (n = 9 tumors/group). Compounds were administered intraperitoneally (i.p.) on day 0 at a dose of 1 mg/kg 1 (<12% EtOH in PBS), 20 mg/kg paclitaxel (<6% EtOH, <6% Kolliphor), or the vehicle for 1. On day 4 animals received a second and final dose of 0.7 mg/kg 1, 20 mg/kg of paclitaxel or the vehicle for 1. Mice were checked for signs of toxicity, weighed and tumors measured by calipers every 2 days for a total of 12 days when vehicle treated tumors began to reach endpoint criterion (1500 mm^3^). Upon completion of the trial, tumors were excised and weighed using a LE225D analytical balance (Sartorius, Goettingen, Germany). All mice were housed in a LAR approved facility at The University of Texas Health Science Center at San Antonio under the IACUC protocol 20170208AR.

## 3. Results

### 3.1. Identification of Dual Immunogenic and TNBC Subtype Selective Activities of Yuanhuacine

In addition to our continued efforts to identify natural products that are selectively cytotoxic to cells representing molecularly diverse TNBC subtypes, we introduced a new assay to identify extracts with immunomodulatory potential (Appendix A). This assay utilizes the THP-1 human monocytic cell line and capitalizes on the fact that these suspension cells become adherent when they differentiate into macrophages and/or dendritic cells as a readout of immune activation. This dual screening approach led to the identification of the NCI plant extract, N117965, which had selective cytotoxic activity against HCC1806 (IC_50_ = 0.4 µg/mL) and HCC70 (IC_50_ = 2.7 µg/mL) cells, which both represent the basal-like 2 (BL2) TNBC subtype, as compared to cells representing each of the other TNBC subtypes (IC_50 s_ > 20 µg/mL) (Figure 1A). Intriguingly, this extract also promoted the differentiation of THP-1 cells with an EC_50_ of 1.8 µg/mL, suggesting that both BL2 selective and immunogenic components with the same range of potency were present in this crude extract (Figure 1B).

To characterize the bioactive component(s) of the crude extract, we performed bioassay-guided fractionation following both TNBC subtype selectivity and THP-1 differentiation activity. These activities co-fractionated and led to the purification of the daphnane type diterpenoid yuanhuacine (**1**) as a primary bioactive component (Figure 2). Consistent with the activity of the crude extract, yuanhuacine (**1**) showed selective cytotoxicity against cell lines representing the BL2 subtype with IC_50_ values of 1.6 and 9.4 nM in HCC1806 and HCC70 cell lines, respectively, and no effect in any of the other TNBC subtypes at concentrations up to 3 µM (Figure 1C and Table 1). Yuanhuacine (**1**) also induced THP-1 differentiation with an EC_50_ of 1.4 nM (Figure 1D and Table 1) demonstrating that this compound retained that same dual BL2 subtype selective and monocyte differentiating activity seen in the crude extract.

### 3.2. The Epoxide in Daphnane Type Diterpenoids Is an Important Pharmacophore for Bioactivity

Daphnane type diterpenoids are commonly associated with plants that belong to *Thymelaeaceae*, including the flower buds of the traditional Chinese medicinal herb *Daphne genkwa* [24]. To gain a deeper understanding of the structure-activity relationship of the daphnane diterpenoid skeleton with regard to both BL2 subtype selectivity and THP-1 differentiation, we purified additional daphnane type diterpenoids (compounds **2**–**8**) from dried flower buds of *Daphne genkwa* (Figure 2). The absence of the C-12 benzoyloxy group (compounds **7** and **8**) or substitution with an acyloxy group (compounds **2** and **3**) did not change the potency against BL2 cells or THP-1 differentiation when compared to yuanhuacine (**1**) (Table 2, Appendix A).

Likewise replacing the C-1′ alkyl group of 1 with an aromatic group (compound **4**) or saturation of the C-1,2 alkene (compounds **6**–**8**) did not alter either cytotoxicity against BL2 TNBC cells or THP-1 differentiating activity (Table 2, Appendix A). However, a lack of the C-6,7-epoxy (compound **5**) substantially decreased both the potency against BL2 cells as well as the differentiation of THP-1 by approximately 100-fold when compared to yuanhuacine (**1**) (Table 2, Appendix A). Together, these studies demonstrate that the C-6,7 epoxide in these daphnane type diterpenoids is an important pharmacophore to induce potent TNBC BL2 selectivity and THP-1 differentiation and that these two phenotypes are closely linked.

### 3.3. Protein Kinase C Agonists Can Induce Selective Cytotoxicity against BL2 TNBC Cells and Promote THP-1 Differentiation

We next sought to determine the mechanism of action of the BL2 selectivity and THP-1 differentiation activities elicited by 1. Daphnane type diterpenoids have been reported to have various mechanisms of action ranging from inhibition of topoisomerase I [25], DNA damage [26], generation of mitochondrial reactive oxygen species (ROS) [27] and activation of protein kinase C (PKC) [28]. To determine whether any of these mechanisms could underlie the BL2 subtype selectivity observed for 1, we evaluated the TNBC subtype selectivity profile of compounds representing each of the aforementioned mechanisms of action. The topoisomerase inhibitors etoposide and the active metabolite of irinotecan, SN-38, were most potent against the HCC1806 and BT-549 cells representing the BL2 and M subtypes, respectively. However, the other BL2 cell line, HCC70, was the most resistant line (Figure 3A,B), suggesting that topoisomerase inhibition is not responsible for the shared BL2 selectivity observed for 1. Similarly, DNA damage induced by gemcitabine (Figure 3C) or the production of ROS [29] by the mitochondrial oxidative phosphorylation uncoupler CCCP (Figure 3D) did not promote BL2 subtype selectivity. In contrast, the tigliane diterpenoid and PKC agonist phorbol 12-myristate 13-acetate (PMA) exhibited BL2 selectivity strikingly similar to that of yuanhuacine (**1**) (Figure 1C and Figure 3E), suggesting that PKC activation could be a previously unrecognized molecular liability of the BL2 subtype of TNBC. Additionally, PMA was the only compound that promoted the differentiation of THP-1 cells with potency in the same range as its activity against BL2 cells (Figure 3F) further supporting that these activities are tightly linked. Altogether, these data suggest that modulation of PKC might be involved in mediating both the BL2 selective cytotoxicity and THP-1 differentiating activities of 1.

### 3.4. Yuanhuacine Cytotoxicity against BL2 TNBC Is Mediated by Protein Kinase C

The protein kinase C family is comprised of several isoforms subdivided into three categories known as conventional (PKCα, βI, βII and γ), novel (PKCδ, ε, η and θ) and atypical (PKCζ and λ/ι) classes [30]. To interrogate whether PKC activation was required for the BL2 subtype selectivity of yuanhuacine (**1**), we pretreated HCC1806 and HCC70 cells with a non-toxic concentration of the PKCα, βI, βII, γ and ε inhibitor, Ro 31-8220, [31] (Appendix A) prior to the addition of 1 at a concentration that reduces viability by 80%. Inhibition of PKC activity with Ro 31-8220 attenuated the cytotoxic effects of 1 in both HCC1806 (Figure 4A) and HCC70 (Figure 4B) cells. However, Ro 31-8220 did not affect the differentiation of THP-1 cells in response to yuanhuacine (**1**), implying that different subsets of PKC isoforms could mediate these effects (Figure 4C). To explore this possibility, we treated TNBC cells with the macrocyclic lactone bryostatin 1, a PKCδ agonist [32]. Unlike yuanhuacine (**1**) or PMA, bryostatin 1 showed no evidence of cytotoxicity to any of the TNBC cell lines, including those of the BL2 subtype, up to a concentration of 1 µM (Figure 4D). However, bryostatin 1 was able to promote THP-1 differentiation with an EC_50_ of 0.5 nM (Figure 4E). Together, these findings demonstrate that although activation of PKC by 1 is required for BL2 subtype selectivity and associated with inducing THP-1 differentiation, the ability of these two phenotypes to be uncoupled with pharmacological agonists and antagonists that have distinct isoform specificity shows that the BL2 selectivity and THP-1 differentiation can be uncoupled by targeting distinct PKC isoforms.

### 3.5. Yuanhuacine Promotes the Expression of Antitumor Cytokines

While the BL2 TNBC subtype selectivity of yuanhuacine (**1**) is novel, previous work has shown that this compound can induce the expression of the antitumor cytokine IFNγ in natural killer cells through activation of NF-κB-mediated transcription [33]. Thus, we examined whether the differentiation of THP-1 cells by 1 was associated with the expression of antitumor cytokines. Indeed, when THP-1 cells were treated with 1 at the EC_75_ for differentiation, 2 nM, for 24 h, we observed an increased transcriptional expression of the antitumor cytokines IFNγ and IL-12 (Figure 5A,B). Pharmacological inhibition of the NF-κB pathway with TPCA-1 or PKC with Ro 31-8220 attenuated expression of these cytokines (Figure 5A,B), suggesting that IFNγ and IL-12 induction by 1 is driven by PKCα, βI, βII, γ or ε isoforms in an NF-κB dependent manner. In contrast to the upregulation of these antitumor cytokines, yuanhuacine (**1**) downregulated the expression of the immunosuppressive cytokine IL-10 (Figure 5C) further supporting the notion that 1 can promote an antitumor-associated immunological phenotype in monocyte-derived cells. Similar to the effects in human cells, murine RAW 264.7 macrophage-derived cells upregulated the expression of IL-12 in response to yuanhuacine (**1**) (Figure 5D). Together, these data show that yuanhuacine (**1**) can promote the expression of antitumor cytokines in immune cells across species.

### 3.6. Yuanhuacine Exhibits Potent In Vivo Antitumor Efficacy against BL2 Tumors

The potent and selective activity of yuanhuacine (**1**) against cells representing the BL2 subtype of TNBC along with the ability to promote antitumor immune signatures in both human and murine monocytic cells prompted us to evaluate the antitumor efficacy of yuanhuacine (**1**) in vivo. We established bilateral subcutaneous HCC1806 xenograft tumors on the flanks of athymic nude mice. Each cohort was composed of five animals distributed to have a comparable initial tumor size of approximately 100 mm^3^. These cohorts were then dosed with vehicle control, 0.7–1 mg/kg yuanhuacine (**1**) or 20 mg/kg paclitaxel by i.p. injection on days 0 and 4 and tumor volume and weight were measured every two days for 12 days (Figure 6A). One yuanhuacine-treated animal died on day 2 after an initial dose of 1 mg/kg, which is why the second dose was reduced to 0.7 mg/kg. Less weight loss was observed with this lower dose (Figure 6D) and the animal that succumbed was excluded from the trial. Yuanhuacine (**1**) demonstrated a significant antitumor response as compared to vehicle-treated animals throughout the duration of the trial as determined by 2-way ANOVA (Figure 6B). In contrast, this analysis showed no significant difference between paclitaxel-treated animals and either other groups over the course of the trial (Figure 6B). However, when tumors were removed and weighed at the end of the trial, a 1-way ANOVA demonstrated significance difference between both 1 and paclitaxel-treated groups as compared to the vehicle group with no significant difference detected between 1 and paclitaxel (Figure 6C). Thus, these data demonstrate that the potent and selective activity of yuanhuacine (**1**) against cells representing the BL2 subtype of TNBC is accompanied by in vivo antitumor efficacy against the HCC1806 xenograft that represents this model even in immunocompromised animals that cannot mount a complete adaptive immune response to the tumor.

## 4. Discussion

TNBC patients do not currently benefit from ER, PR and HER2 directed therapies, which have revolutionized the treatment of other forms of breast cancer. The molecular profiling of TNBC tumors and cell lines has allowed this disease to be subtyped by shared driver pathways that could be targeted with new or repurposed drugs. TNBCs also have a higher degree of immunogenicity than other breast cancers, which has prompted the study of immune checkpoint inhibitors in these patients, particularly when the tumor has a favorable immune microenvironment. In our drug discovery efforts, we focused on identifying natural products that promoted both selective cytotoxicity against cell lines representing distinct molecular TNBC subtypes while also inducing an immunological signature associated with antitumor immunity. Herein, we identified several potent daphnane type diterpenoids (compounds **1**–**8**), including yuanhuacine (**1**), that demonstrates potent and selective cytotoxicity against cells representing the BL2 subtype of TNBC.

Traditional Chinese medicinal herbs and their active constituents have long been evaluated for the treatment of carcinomas, including TNBCs, due to their ability to target multiple oncogenic signaling pathways [34]. Yuanhuacine (**1**), one of the constituents of the medicinal plant *Daphne genkwa*, has been previously reported to have in vitro cytotoxic activity against over a dozen diverse human cancer cell lines at concentrations ranging from 1–27 µM [21,35,36,37]. This is in striking contrast to the single digit nanomolar potency we observed for this compound in cell lines that represent the BL2 subtype of TNBC; HCC1806 (IC_50_: 1.6 nM) and HCC70 (IC_50_: 9.4 nM). The only previously published data describing the in vivo antitumor efficacy of yuanhuacine (**1**) showed a very modest decrease in growth of a H1993 non-small cell lung cancer xenograft in animals with daily oral dosing of 1 mg/kg for a total dose of 21 mg/kg [36]. We speculate that the lack of in vivo antitumor studies with systemic administration in the literature could be due to a low therapeutic index against models that require micromolar exposure levels to inhibit tumor growth. In contrast, we demonstrated robust in vivo antitumor efficacy against the sensitive HCC1806 xenograft with a total dose of only 1.7 mg/kg administered by i.p injection. It is worth highlighting that this suppression of HCC1806 tumor growth by 1 occurred in an immunocompromised model. We anticipate an even more pronounced antitumor effect might be observed in an immunocompetent syngeneic model bearing BL2-like murine tumors due to the combined BL2-selective cytotoxic and immunological effects of 1. However, there are currently no murine TNBC models that have been molecularly subtyped to this extent. Ongoing experiments to characterize the exact molecular liability that sensitizes human BL2 TNBC cell lines to 1 will allow us to modify murine TNBC lines to optimize their sensitivity to yuanhuacine and facilitate these future studies. For now, these data demonstrate that yuanhuacine exploits a somewhat unique molecular liability of BL2 cells, which allows for antitumor efficacy with an acceptable therapeutic index.

Like many natural products, several mechanisms of action have been attributed to yuanhuacine (**1**) that have been proposed to underlie its ability to inhibit the growth of cancer cells. We took advantage of the diverse molecular liabilities of TNBC cell lines to compare the relative selectivity of compounds that target different putative mechanisms of 1 to determine which of them could underlie the BL2 selectivity we observed. This strategy successfully identified PKC activation as the mechanism of BL2 selectivity of 1 demonstrating that agents that can target PKC might have an optimal therapeutic index in selectively targeting the BL2 subtype of TNBC at concentrations that are not toxic to surrounding benign tissue such as human mammary epithelial cells (HMEC) where 1 has an IC_50_ of 14.0 µM [38]. While not anticipated, it is reasonable for PKC activation to be a molecular liability of the BL2 subtype of TNBC, which is characterized in part by a dependency on glycolytic pathways [8], as some PKC isoforms can directly phosphorylate the insulin receptor substrate 1 (IRS1) leading to starvation of cells with a high glucose dependency [39,40]. The comparative pharmacological approach we used to interrogate the mechanism of action of 1 is similar to the utilization of cancer dependency databases, such as Project Achilles [41], that can facilitate the identification of the mechanism of action of a compound based on its profile of selectivity among cell lines with distinct genetic liabilities [13]. These are valuable tools when uncovering the mechanism of action of a particular feature (such as BL2 TNBC subtype selectivity) of natural products like yuanhuacine that have a long and diverse pharmacological history.

One important finding in our study is the elucidation that the C-6,7-epoxy of the yuanhuacine chemotype substantially contributes to the potency of both the cytotoxicity against BL2 cells as well as the differentiation of THP-1 cells. Although alpha epoxidation at these carbons is common in many daphnane type diterpenoids (e.g., yuanhuacine, mezerein, etc.), they are rarely found in the tigliane diterpene family (e.g., PMA, ingenol mebutate, etc.) [42]. The synthetic C-6,7-epoxytigliane tigilanol tiglate (EBC-46) was recently approved in the EU for the localized treatment of non-metastatic skin cancers in dogs. Interestingly, EBC-46 displays a distinct PKC isoform activation pattern from non-epoxidated tigliane diterpenes marked by a preference for PKC-βI/II isoforms and, to a lesser degree, PKCα/γ [43,44]. These are all PKC isoforms that are inhibited by Ro 31-8220, which attenuated the ability of yuanhuacine (**1**) to inhibit BL2 TNBC cells in our study. Therefore, we hypothesize that the mechanism of action by which 1 activates PKC to selectively inhibit the BL2 subtype of TNBC could be shared by EBC-46. A first-in-human phase I study of EBC-46 administered by intratumoral injection demonstrated a promising safety and efficacy profile [45]. Additionally, a phase I/II clinical trial of intratumoral administration of tigilanol tiglate in combination with the immune checkpoint inhibitor prembolizumab in stage 3/4 melanoma was recently initiated (NCT04834973) [46]. Our data suggest that C-6,7-epoxy PKC agonists like 1 or EBC-46 have the potential for the treatment of patients with the BL2 subtype of TNBC by systemic administration, either alone or in combination with immune checkpoint inhibition, due to a unique liability that makes them exquisitely sensitive to this class of compounds while also promoting an antitumor associated immunological response.

## 5. Conclusions

The search for new targets and drug leads for the treatment of TNBC led us to identify yuanhuacine (**1**) and related daphnane diterpenes as potent PKC agonists with BL2 TNBC selectivity and immunogenic activity that have in vivo antitumor efficacy.

## Figures and Tables

**Figure 1 cancers-13-02834-f001:**
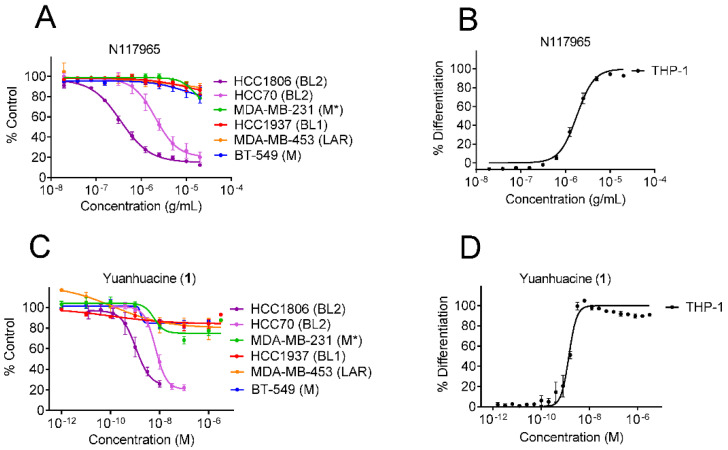
Yuanhuacine selectively inhibits the growth of cell lines that represent the BL2 TNBC subtype and promotes THP-1 differentiation. (**A**) Concentration response curves of the growth of cells that represent diverse TNBC molecular subtypes when treated with crude N117965 extract for 48 h. * The MDA-MB-231 cell line was originally classified as the MSL subtype and regrouped into the consolidated M subtype due to its mesenchymal phenotype. (**B**) Concentration response curve of THP-1 cell differentiation when treated with crude N117965 extract for 24 h. (**C**) Concentration response curves of the growth of cells that represent diverse TNBC molecular subtypes when treated with 1 for 48 h. (**D**) Concentration response curve of THP-1 cell differentiation when treated with 1 for 24 h. Error bars indicate the SEM of 2–4 biological replicates.

**Figure 2 cancers-13-02834-f002:**
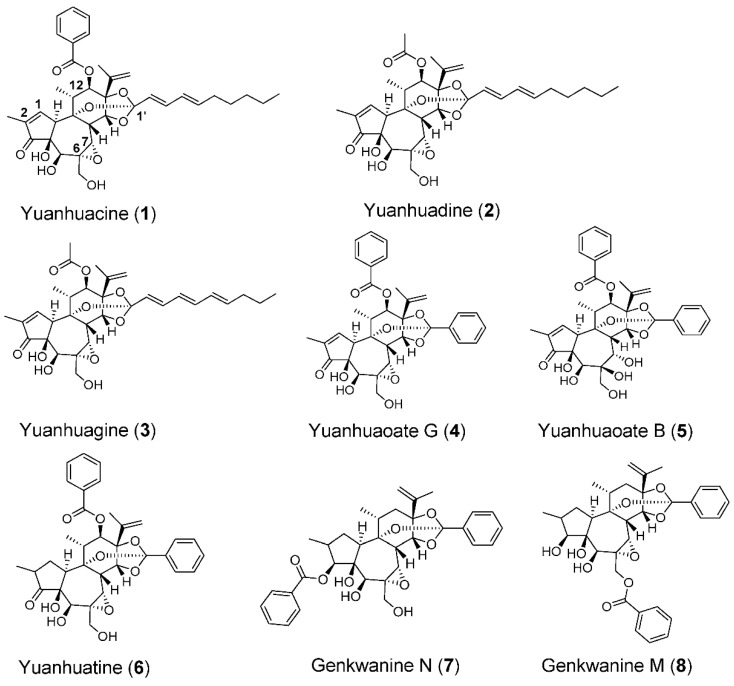
Structures of daphnane type diterpenoids identified in this study.

**Figure 3 cancers-13-02834-f003:**
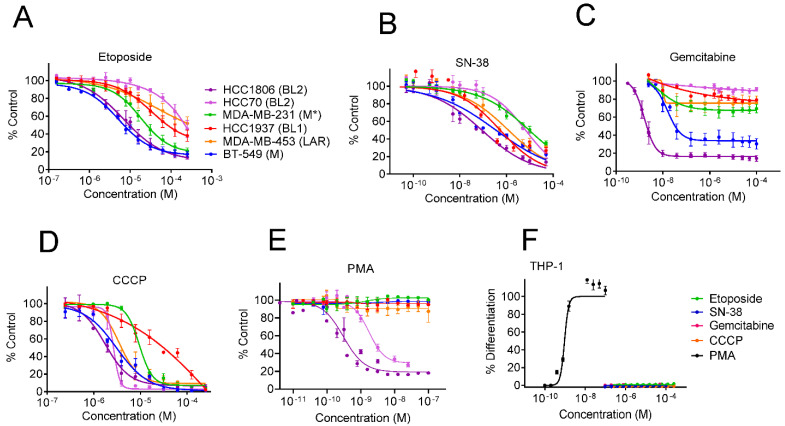
TNBC selectivity and THP-1 differentiation profiles of compounds with mechanisms associated with yuanhuacine. Concentration response curves of the growth of cell lines that represent diverse TNBC molecular subtypes when treated with the topoisomerase inhibitors (**A**) etoposide and (**B**) SN-38, (**C**) the DNA damaging agent gemcitabine, (**D**) the mitochondrial uncoupling agent CCCP, or (**E**) the PKC agonist PMA for 48 h. (**F**) Concentration response curve of THP-1 cell differentiation when treated with etoposide, SN-38, gemcitabine, CCCP, or PMA for 24 h. * The MDA-MB-231 cell line was originally classified as the MSL subtype and regrouped into the consolidated M subtype due to its mesenchymal phenotype. Error bars indicate the SEM of 2–4 biological replicates.

**Figure 4 cancers-13-02834-f004:**
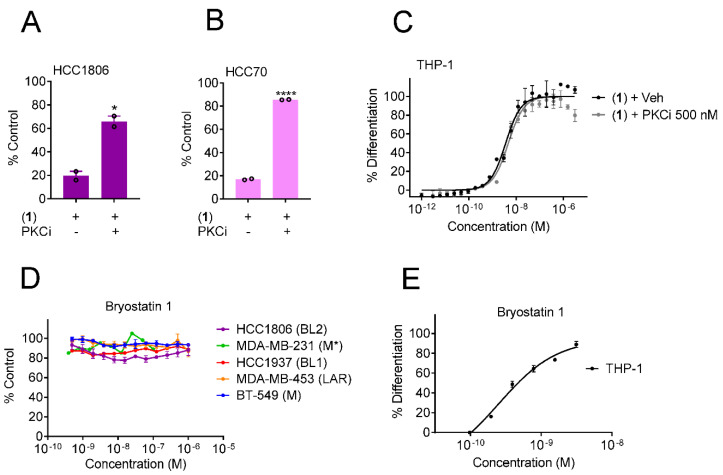
BL2 subtype selectivity of yuanhuacine is dependent on PKC activity. Growth of (**A**) HCC1806 or (**B**) HCC70 cells pretreated with 500 nM of Ro-31-8220 (PKCi) for 4 h and then treated with a concentration of 1 that results in 80% growth inhibition on its own (50 or 200 nM, respectively) for 48 h. Significance was determined by unpaired *t*-test. * *p* < 0.05, **** *p* < 0.0001. (**C**) Concentration response curve of THP-1 differentiation when cells were pretreated with 500 nM of Ro-31-8220 (PKCi) for 4 h and then treated with 1 for 24 h. (**D**) Concentration response curves of the growth of cells representing diverse TNBC molecular subtypes when treated with bryostatin 1. (**E**) Concentration response curve of THP-1 differentiation when cells were treated with bryostatin 1. * The MDA-MB-231 cell line was originally classified as the MSL subtype and regrouped into the consolidated M subtype due to its mesenchymal phenotype. Error bars indicate the SEM of 2–4 biological replicates.

**Figure 5 cancers-13-02834-f005:**
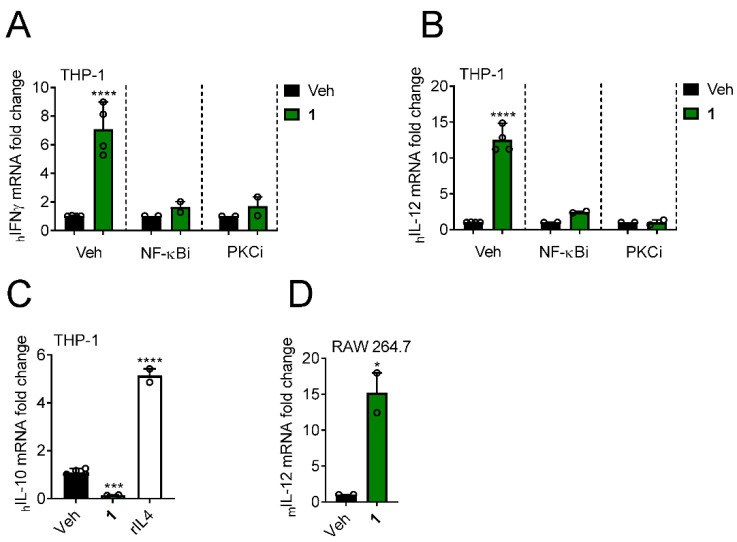
Yuanhuacine induces NF-κB dependent expression of antitumor cytokines. Expression of (**A**) IFNγ and (**B**) IL-12 mRNA in THP-1 cells pretreated with 1 µM of the indicated inhibitor or vehicle for 4 h and then treated with the 2 nM 1 or vehicle for 24 h. Significance determined by 2-way ANOVA with Šídák’s posthoc test. **** *p* < 0.0001. (**C**) Expression of IL-10 mRNA in THP-1 cells treated with 2 nM 1 or 1 µg/mL of recombinant human IL-4 for 24 h. *** *p* < 0.001. (**D**) IL-12 mRNA expression in RAW 264.7 cells treated with 2 nM 1 for 24 h. Significance determined by unpaired *t*-test. * *p* < 0.05.

**Figure 6 cancers-13-02834-f006:**
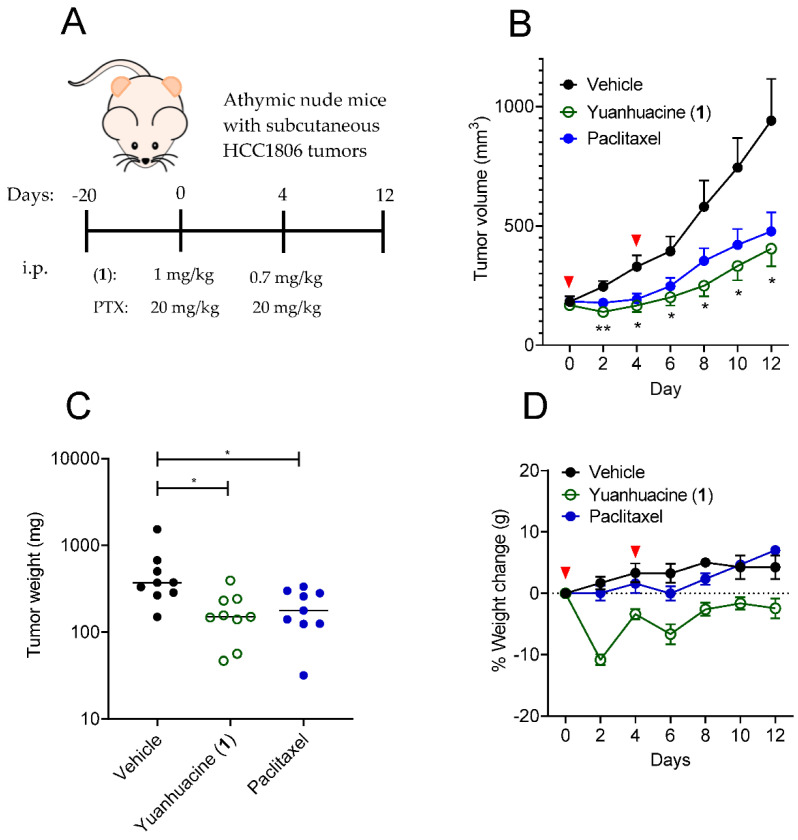
Yuanhuacine exhibits antitumor activity against a xenograft model of the BL2 subtype of TNBC. (**A**) Timeline of the antitumor trial. (**B**) Growth of HCC1806 flank xenograft tumors in athymic mice treated by i.p. injection with 1 (1 mg/kg day 0 and 0.7 mg/kg day 4), PTX (20 mg/kg days 0 and 4) or vehicle. Red arrows indicate days animals were dosed. Significance determined by 2-way ANOVA with Tukey’s posthoc test. * *p* < 0.05, ** *p* < 0.01. (**C**) The weight of tumors collected at day 12. Significance determined by 1-way ANOVA with Dunnett’s posthoc test. * *p* < 0.05. (**D**) Percent change in weight during the course of the trial. Red arrows indicate days animals were dosed.

**Table 1 cancers-13-02834-t001:** Potency of 1 for the inhibition of TBNC cell growth (IC_50_) or the differentiation of THP-1 cells (EC_50_).

Cells	IC_50_ ± SEM (nM)	EC_50_ ± SEM (nM)
HCC1806 (BL2)	1.6 ± 0.4	-
HCC70 (BL2)	9.4 ± 1.6	-
MDA-MB-231 (M)	>3000	-
HCC1937 (BL1)	>3000	-
MDA-MB-453 (LAR)	>3000	-
BT-549 (M)	>3000	-
THP-1	-	1.4 ± 0.2

**Table 2 cancers-13-02834-t002:** Potency of **1**–**8** for growth inhibition of BL2 subtype TBNC cells (IC_50_) or differentiation of THP-1 cells (EC_50_).

Compound	HCC1806 IC_50_ ± SEM (nM)	HCC70 IC_50_ ± SEM (nM)	THP-1 EC_50_ ± SEM (nM)
**1**	1.6 ± 0.4	9 ± 1	1.4 ± 0.2
**2**	1.3 ± 0.4	3.8 ± 0.4	1.4 ± 0.1
**3**	3.8 ± 0.7	12 ± 1	2.2 ± 0.2
**4**	2.7 ± 0.6	15 ± 1	2.2 ± 0.4
**5**	149 ± 31	1612 ± 472	149 ± 1
**6**	7 ± 1	22 ± 2	8.6 ± 0.4
**7**	1.0 ± 0.1	6 ± 1	1.3 ± 0.1
**8**	1.3 ± 0.2	10 ± 1	0.7 ± 0.1

## Data Availability

Data is contained within the article or Appendix A.

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
