# Peer review of "Yuanhuacine Is a Potent and Selective Inhibitor of the Basal-Like 2 Subtype of Triple Negative Breast Cancer with Immunogenic Potential"

_cancers, 2021, doi:10.3390/cancers13112834_

Round 1

Reviewer 1 Report

The manuscript describes the discovery and mechanistic characterization of breast cancer subtype-specific natural product inhibitor.  The compound yuanhuacine was identified through a clever dual screen for differentiation of THP-1 monocytes as indicator of immune activation and cytotoxicity against breast cancer subtypes. Yuanhuacine had exquisite potency in the immune activation assay and the IC50 matched that of cytotoxicity against HCC1806 cells (BL2/basal-like 2 subtype). The selectivity over other subtypes was impressive and convincing, suggesting that a specific molecular liability in BL2 subtypes is targeted by the natural product. SAR studies provided evidence for the epoxide to be a critical pharmacophore. Experiments with pan- and isoform-specific PKC inhibitors revealed that PKC activation is the mechanism of action in the BL2 subtype, while other known activities of related diterpenes were found not to be responsible for the phenotypes, based on comparison with etoposide, gemcitabine and other agents.  Yuanhuacine showed activity in immunocompromised mice (HCC1806 xenograft).

This is a comprehensive manuscript and reveals PKC activation as a promising avenue to target BL2 subtypes of breast cancer.

Comments:

  • It is interesting that the compound showed activity in immunocompromised mice. I would expect that a larger effect would be seen in immunocompetent mice through the PKC-mediated mechanism. I understand that the authors wanted to stay with HCC1806 and had to use immunocompromised mice for the xenograft. Could the authors comment if a similar study may be done in immunocompetent mice in the future?
  • The authors should indicate assay time points in all figure legends (missing in Figures 1 and 3)

Reviewer 2 Report

This is a very well conducted and interesting study. The results are presented and discussed in a clear and incisive way. In my opinion, the manuscript can be published as it is, taking into account small suggestions as follows:

-Lines 181-183: I would be more cautious in saying that Yuanhuacine was responsible for both the BL2 subtype selectivity as well as the monocyte differentiating activity in the extract. It is possible that other compounds with the same properties could be present in the extract

-Figure 3A: is this figure necessary?

-In the legend of Figure 5 the (D) is missing
